# A new inference approach for training shallow and deep generalized linear models of noisy interacting neurons

**Gabriel Mahuas**[*]
Laboratoire de physique de l'École normale supérieure, 24 rue Lhomond, 75005, Paris, France
PSL University, CNRS, Sorbonne Université, Université de Paris,
Institute of Science and Technology Austria
Am Campus 1, A-3400 Klosterneuburg, Austria

**Giulio Isacchini**[†]
Laboratoire de physique de l'École normale supérieure, 24 rue Lhomond, 75005, Paris, France
PSL University, CNRS, Sorbonne Université, Université de Paris,

**Olivier Marre**
Institut de la Vision
Sorbonne Université, INSERM, CNRS
17 rue Moreau, F-75012, Paris, France

**Ulisse Ferrari**[‡]
Institut de la Vision
Sorbonne Université, INSERM, CNRS
17 rue Moreau, F-75012, Paris, France
ulisse.ferrari@gmail.com

**Thierry Mora**[‡]
Laboratoire de physique de l'École normale supérieure, 24 rue Lhomond, 75005, Paris, France
PSL University, CNRS, Sorbonne Université, Université de Paris,
thierry.mora@gmail.com

## Abstract

Generalized linear models are one of the most efficient paradigms for predicting the correlated stochastic activity of neuronal networks in response to external stimuli, with applications in many brain areas. However, when dealing with complex stimuli, the inferred coupling parameters often do not generalize across different stimulus statistics, leading to degraded performance and blowup instabilities. Here, we develop a two-step inference strategy that allows us to train robust generalized linear models of interacting neurons, by explicitly separating the effects of correlations in the stimulus from network interactions in each training step. Applying this approach to the responses of retinal ganglion cells to complex visual stimuli, we show that, compared to classical methods, the models trained in this way exhibit improved performance, are more stable, yield robust interaction networks, and generalize well across complex visual statistics. The method can be extended to deep convolutional neural networks, leading to models with high predictive accuracy for both the neuron firing rates and their correlations.

[*]Current address: Sorbonne Université, INSERM, CNRS, Institut de la Vision, 17 rue Moreau, F-75012, Paris, France
[†]Current address: Max Planck Institute for Dynamics and Self-organization, 37077 Göttingen, Germany
[‡]These authors contributed equally

# 1 Introduction

The pioneering work of J.W. Pillow and colleagues [1] showed how the Generalized Linear Model (GLM) can be used for predicting the stochastic response of neurons to external stimuli. Thanks to its versatility [2], high performance, and easy inference, the GLM has become one of the reference models in computational neuroscience. Nowadays, its applications range from retinal ganglion cells [1], to neurons in the LGN [3], visual [4], motor [5], parietal [6] cortices, as well as other brain regions [7, 8, 9]. However, the GLM has also shown some significant limitations that has prevented its application to an even wider spectrum of contexts. In particular, the GLM shows unsatisfying performance when applied to the response to complex stimuli with spatio-temporal correlations much stronger than white noise, as for example naturalistic images [10] or videos [11].

A first limitation is that the inferred parameters depend on the stimulus used for training. This happens not only for the part of the model that deals with the external stimulus, which typically suffers a change in the stimulus statistics, but also for the couplings parameters quantifying interactions between the neurons of the network. However, if these couplings are to reflect an underlying network of biological interactions, they should be stimulus independent. In addition, and as we show in this paper, this lack of generalizability comes with errors in the prediction of correlated noise between neurons. This issue can strongly limit the application of GLM for unveiling direct synaptic connections between the recorded neurons [12] and for estimating the impact of noise correlations in information transmission [1].

A second issue is that the GLM can be subject to uncontrollable and unnatural self-excitation transients [13, 14, 15]. During these strong and positive feedback loops, the network's past activity may drive its current state to excitations above naturalistic levels, in turn activating neurons in subsequent time steps and resulting in a transient of very high, unrealistic activity. This problem limits the use of the GLM as a generative model—it is often necessary to remove those self-excitation runs by hand. Ref. [13] proposed an extension of the GLM that also includes quadratic terms limiting self-excitations of the network, but this comes at the price of more fitting parameters and harder inference. Ref. [14] showed that a GLM that predicts the responses several time-steps ahead in time [16] limits self-excitation, but this implies higher computational complexity and the risk of missing fine temporal structures. Alternatively, Ref. [15] proposed an approximation to estimate the stability of the inferred GLM model, and then used a stability criterion to constrain the parameter space over stable models. However the resulting models are sub-optimal, with degraded performance.

Thirdly, because neuronal responses are highly non-linear and hard to model for complex stimuli, the GLM fails to predict those responses correctly, even for early visual areas such as the retina [11]. Recently deep convolutional neural networks (CNNs) have been shown to outperform the GLM at predicting individual neuron mean responses [10, 17, 18, 19]. Compared to the GLM, these deep CNNs benefit from a more flexible and richer network architecture allowing for strong performance improvements [10]. However, the GLM retains an advantage over CNNs: thanks to the couplings between neurons in the same layer, it can account for both shared noise across the population and self-inhibition due to refractoriness. This feature, which is missing from deep CNNs [10], can be used to study how noise correlated in space and time impacts the population response [1]. A joint model combining the benefits of the deep architecture of CNNs and the neuronal couplings of the GLM is still lacking. It would allow for a more detailed description of the neuronal response to stimulus.

In this paper we develop a two-step inference strategy for the GLM that solves these three issues. We apply it to recordings in the rat retina subject to different visual stimulations. The main idea is to use the responses to a repeated stimulus to infer the GLM couplings without including the stimulus processing component. Then, in a second, independent step, we infer the parameters of the model pertaining to stimulus processing. Our approach allows for a wide variety of architectures, including deep CNNs. Finally, we introduce an approximation scheme to put together the two inference results into a single model that can predict the joint network response from the stimulus.

All codes and data for the algorithms presented in this paper are available at https://github.com/gmahuas/2stepGLM

## 2 Recordings

Retinal ganglion cells of a long-evans rat were recorded through a multi-electrode array experiment [20, 21] and spike-sorted with *SpyKING CIRCUS* [22]. Cell activity was stimulated with one unrepeated and two repeated videos of checkerboard (white-noise) and moving bars. For the checkerboard, we used the unrepeated ($1350s$) and one of the two repeated videos ($996s$ in total for 120 repetitions) for training, and the second repeated video for testing ($756s$ in total for 120 repetitions). Similarly, for the moving bars video we used the unrepeated ($1750s$) and one of the two repeated videos ($165s$ in total for 50 repetitions) for training, and the second repeated video for testing ($330s$ in total for 50 repetitions). In addition, we also recorded responses from a full-field movie with naturalistic statistics [21].

After sorting, we applied a spike-triggered average analysis to locate the receptive fields of each cell. Then, we used the response to full-field stimulation to cluster cells into different cell-types. In this work we focus on a population of $N = 25$ OFF Alpha retinal ganglion cells, which tile the visual field through a regular mosaic. The responses to both checkerboard and moving bars stimulations showed strong correlations, which we decompose into the sum of stimulus and noise correlations. Stimulus correlations are correlations between the cell mean responses (Peristimulus time histogram or PSTH). They are large only for the bars video, mostly because the video itself has strong and long-ranged correlations. Noise correlations, on the other hand, are due to shared noise from upstream neurons and gap junctions between cells in the same layer [23], and mostly reflect the architecture of the underlying biological network. Consistently, noise correlations were similar in the response of the two stimulations. In Suppl. sect. S1 we present additional statistics of the data.

## 3 Generalized linear model

In our Poisson GLM framework, $n^i(t)$, the number of spikes emitted by cell $i$ in the time-bin $t$ of duration $dt = 1.67ms$, follows a Poisson distribution with mean $\lambda^i(t)$: $n^i(t) \sim \text{Pois}(\lambda^i(t))$. The vector of the cells' firing rate $\{\lambda^i(t)\}_{i=1}^N$, with $N = 25$ is then estimated as

$$\lambda^i(t) = \exp\left\{ h^i_{\text{offset}} + h^i_{\text{stim}}(t) + h^i_{\text{int}}(t) \right\} , \tag{1}$$

where $h^i_{\text{offset}}$ accounts for the cell's baseline firing rate and where

$$h^i_{\text{int}}(t) = \sum_j \sum_{\tau > 0} J_{ij}(\tau) n^j(t - \tau) \tag{2}$$

accounts for both past firing history of cell $i$ itself and the contribution coming from all other cells in the network: $J_{ii}$ are the spike-history filters, whereas $J_{i \neq j}$ are coupling filters. Both integrate the past up to 40ms. $h^i_{\text{stim}}(t)$ is a contribution accounting for stimulus drives, which takes the form of a linear spatio-temporal convolution in the classical GLM:

$$h^i_{\text{stim}}(t) = \sum_{\tau > 0} \sum_{xy} K_{x,y}(\tau) S_{x,y}(t - \tau) , \tag{3}$$

where $S_{x,y}(t)$ is the stimulus movie at time $t$, $\{x, y\}$ being the pixel coordinates and $K_{x,y}(\tau)$ is a linear filter that integrates the past up to 500 ms. Later in the paper, we will go beyond this classical architecture and will allow for deep, non-linear architectures.

In order to regularize couplings and spike-history filters during the inferences, we projected their temporal part over a raised cosine basis [1] of 4 and 7 elements respectively, and added an $L1$-regularization $= 0.1$, which we kept the same for all the inferences. In addition, we imposed an absolute refractory period of $\tau^i_{\text{refr}}$ time-bins (calculated from the training set) during simulations and consequently the $J_{ii}(\tau)$ were set to zero for $\tau \leq \tau^i_{\text{refr}}$. In order to lower its dimension, the temporal behavior of stimulus filter $K_{x,y}(\tau)$ was projected on a raised cosine basis with 10 element. In addition we included an L1 regularization over the basis weights and a L2 regularization over the spatial Laplacian to induce smoothness.

All the inferences were done by log-likelihood (log-$\ell$) maximization with Broyden-Fletcher-Goldfarb-Shanno (BFGS) method, using the empirical past spike activity during training [1]. For easy comparison, all the performances discussed below are summarized in Table 1.

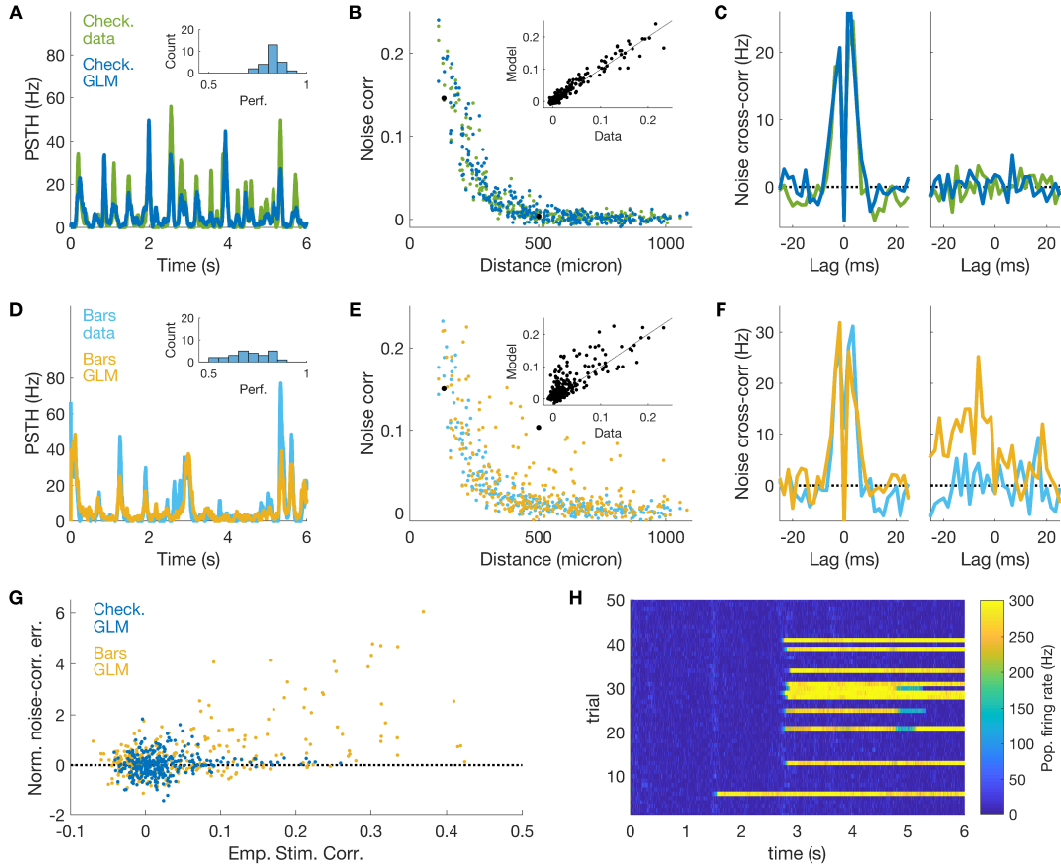

Figure 1: **GLM fails to predict noise correlations in the presence of strong stimulus correlations.** A) PSTH prediction for the response of an example cell to checkerboard stimulation. Inset: histogram of the model performance (Pearson correlation between empirical and model PSTH) for all cells in the population. B) Empirical and model predicted noise correlations versus distance between the cells. Inset: scatterplot. C) Empirical and model predicted noise cross-correlation between a nearby and a distant example cells. D,E,F) same as A,B,C, but for the response to moving bars stimulation. Note that the model overestimates noise correlations between certain pairs of distant cells. G) Error in the prediction of noise correlations normalized over their empirical value versus the empirical value of stimulus correlations. H) Population firing rate in time during model simulations of the responses to the moving bars stimulus. Note the transient of unnatural high activity due to self-excitation within the model.

## 4   Failure of GLM for complex stimuli

We inferred the GLM by whole log-$\ell$ maximization from both the response to the checkerboard and moving bars non-repeated stimulations, and then simulated its response to the repeated videos (Fig. 1). Consistent with [1], in the case of the checkerboard stimulus, the model can predict with high accuracy the PSTH of all cells (Fig. 1A, mean Pearson's $\rho = 0.82 \pm 0.05$ std). It also reproduces the values of the zero-lag (17 ms window) noise correlations for all cell pairs (Fig. 1B, coefficient of determination CoD= $0.94$, computed as $1 - \mathrm{var(error)/var(data)}$), and the temporal structure of noise cross-correlations (Fig. 1C).

The model performance is very degraded for the moving bars video—a stimulus characterised by long-range correlations. The model reproduces the empirical PSTH with rather good accuracy (Fig. 1D, $\rho = 0.71 \pm 0.10$ std) and shows fair overall accuracy on the noise correlations (Fig. 1E, CoD= $0.55$). However it overestimates the value of the noise correlation for certain distant cell pairs (Fig. 1E&F). A closer look reveals that the model overestimates noise correlations for pairs of cells that are strongly stimulus-correlated (Fig. 1G). Here the error in the estimates is normalized over the empirical value

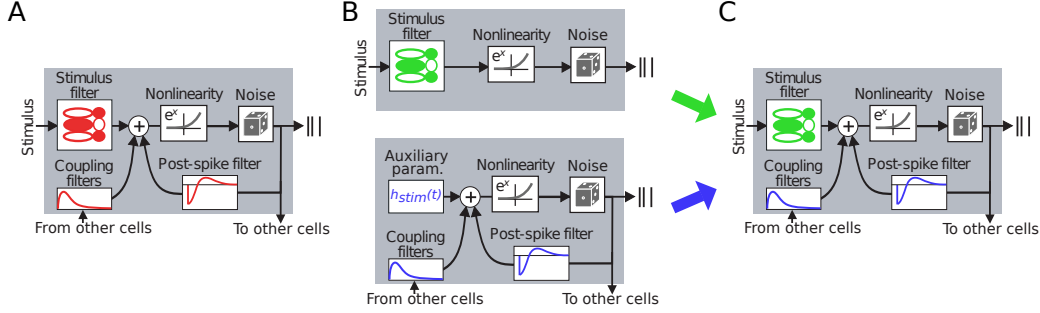

Figure 2: **Two-step inference of couplings and spike-history filters.** A) Whole log-$\ell$ maximization [1] trains couplings and spike-history filters together with the stimulus filter. B) Two-step inference trains couplings filters and stimulus filters by running two independent log-$\ell$ maximizations. Top: we remove coupling filters and infer the equivalent of an LNP model for each cell. Bottom: we run an inference over repeated data where we add auxiliary variables (instead of the stimulus filter) to exactly enforce the PSTH prediction. C) We build together the model by using the previously inferred parameters. A correction needs to be added (not shown, see text).

of the noise correlations with a cut-off at three standard deviations. Interestingly, the effect is strong only for the moving bars video, as stimulus correlations are small for checkerboard stimulation. These results show that the inferred couplings of the GLM do not depend only on the correlated noise among the neurons, but can also be influenced by stimulus correlations. This prevents the inferred couplings from generalizing across stimuli. In addition, we observed several self-excitation transients when simulating the GLM inferred from the moving-bars stimulus ($10\%$ of the time, in $36\%$ of the repetitions, Fig.1H, versus $0\%$ for the model inferred from the checkerboard stimulus). This effect is probably the consequence of the over-estimation of those cell-to-cell couplings in the moving-bars stimulus, which drive the over-excitation of the network.

All these issues can be ascribed to the fact that by maximising the whole log-$\ell$ over all the parameters simultaneously, the GLM mixes the impact of stimulus correlations with neuronal past activity. In the next section we develop an inference strategy that disentangles stimulus from noise correlations and infer their parameters independently.

## 5 A two-step inference approach

In order to disentangle the inference of the couplings between neurons from that of the stimulus filters, we split the model training into two independent steps. We name this approach "two-step" inference (Fig. 2).

*Coupling inference.* We run a log-$\ell$ maximization inference over the response to a repeated video stimulation. Instead of inferring the parameters of a stimulus filter ($K_{x,y}(\tau)$ in Eq. 3), we treat the terms $h^i_{\text{stim}}(t)$ of Eq. 1 as auxiliary parameters that we infer directly from data (Fig. 2B). The log-$\ell$ derivative over these parameters is proportional to the difference between empirical and model-predicted PSTH. As a consequence, thanks to the repeated data, the addition of these parameters allows for enforcing the value of the PSTH exactly when the corresponding log-$\ell$ gradient vanishes. In this way, stimulus correlations are perfectly accounted for, and the couplings only reflect correlated noise between neurons. As for the GLM inferred with whole log-$\ell$ maximization, we imposed an absolute refractory period of $\tau^i_{\text{refr}}$ time-bins and thus set $J_{ii}(\tau)$ to zero for $\tau \leq \tau^i_{\text{refr}}$.

*Filter inference.* We run the inference of a GLM model without the couplings between neurons or with themselves (spike-history filter) using the responses to the unrepeated stimulus (single-neuron linear-nonlinear Poisson (LNP) models [24], Fig. 2B). This inference allows us to predict the mean firing rate $\lambda^i$ of each neuron.

*Full model.* Once couplings and stimulus filters are inferred, we can combine them to build up the full model (Fig. 2C). This cannot be done straightforwardly because the addition of the couplings will change the firing rate prediction of LNP model. As the average contribution of interactions on

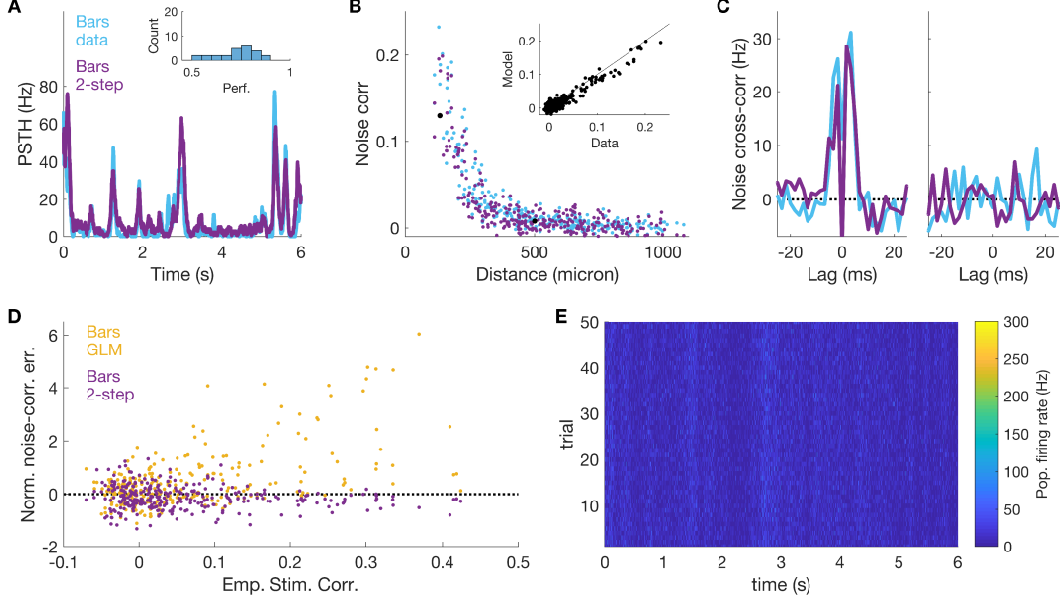

Figure 3: **Two-step inference retrieves noise correlations independently of the strong stimulus correlations in the moving bars video.** A) PSTH prediction for an example cell. Inset: histogram of the model performance for all cells. B) Empirical and model-predicted noise correlations versus distance between the cells. Inset: scatterplot. C) Empirical and model predicted noise cross-correlation between example pairs of nearby and distant cells. D) Normalized error in the prediction of noise correlations plotted versus the empirical value of the stimulus correlations. E) Population activity during model simulation shows no self-excitation transients.

the activity of the cells were not taken into account during the inference of the stimulus model, we need to correct for this effect. To do so, we subtract the mean contribution of the coupling term: $h^i_{\text{int}}(t) \to h^i_{\text{int}}(t) - \langle h^i_{\text{int}}(t) \rangle_{\text{noise}\sim\text{Pois}}$. This correction is equivalent to modify Eq. 2 into

$$\sum_j \sum_{\tau>0} J_{ij}(\tau) n^j(t-\tau) \to \sum_j \sum_{\tau>0} J_{ij}(\tau) \Big( n^j(t-\tau) - \lambda^i(t-\tau) \Big) . \qquad (4)$$

Lastly, in order to account for the addition of absolute refractory periods, we added a term $\sum_{\tau=1}^{\tau^i_{\text{refr}}} \lambda^i(t-\tau)$ for each neurons (Suppl. Sect. S2). To compute all the corrections, we therefore only need the past firing rates $\lambda^i(t)$ of all neurons in the absence of the couplings, which are given by the LNP model predictions. This allows the full model to predict the neuronal response to unseen (testing) data.

Note that this last correction is only an approximation. An exact alternative would be to perform the inference of the GLM stimulus filters as before, but in the presence of coupling filters fixed to the values inferred in the first step. Applying this approach to our data brought no improvement in terms of model accuracy, at the cost of more complex and time-consuming inferences.

We first applied our two-step inference to the response to checkerboard stimulation and obtained very similar results to whole log-$\ell$ maximization (Table 1). By constrast, performance was improved in the case of the moving bars stimulus (Fig. 3). The two inference approaches yielded similar performances for the PSTH (Fig. 3A, $\rho = 0.72 \pm 0.10$ std, versus $\rho = 0.71 \pm 0.10$ std), but for noise correlations we obtained much better results (Fig. 3B, CoD= 0.91, versus CoD= 0.55). In particular, the model avoids the overestimation the noise correlations for distant pairs (Fig. 3B&C) that we obtained with whole log-$\ell$ maximization (Fig. 1E&F). With the two-step inference, the strong stimulus correlations of the moving bars video do not affect the model inference as was the case for whole log-$\ell$ maximization (Fig. 3D). In addition the model is much more stable, and we never observed self-excitation for either stimulus when simulating the model (Fig.3E, versus 10% of the time, Fig. 1H). In Table 1 we report all the performance for the different cases.

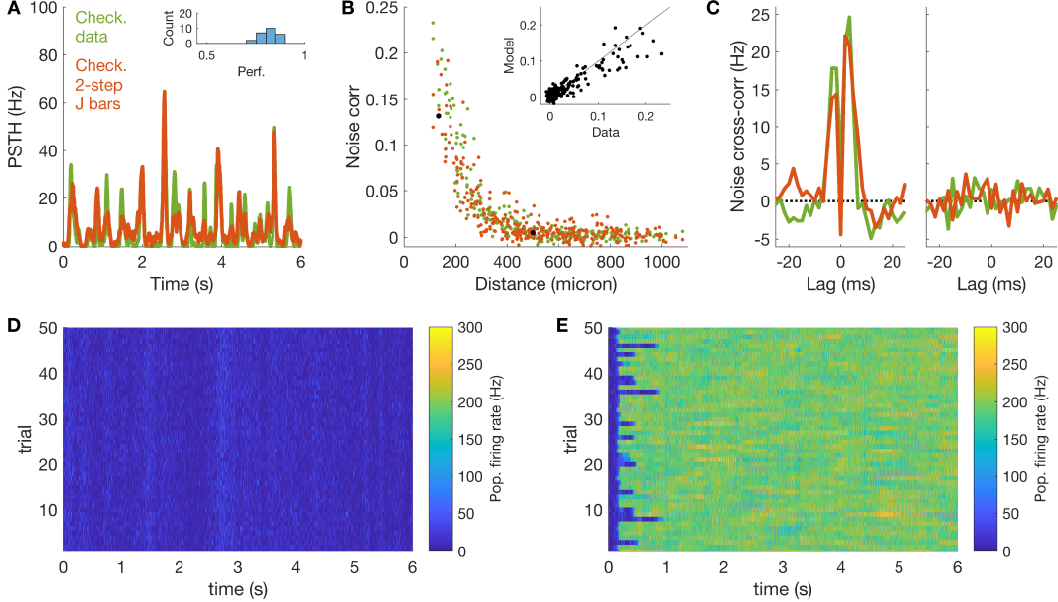

Figure 4: **Two-step inference allows for generalizing across stimulus ensembles** A,B,C,D) Simulation of the checkerboard responses for a model where stimulus filters were inferred from the response to checkerboard, and couplings filter were inferred from the moving bars data with our two-step inference. A) PSTH predictions. B) Noise correlations. C) Noise cross-correlation. D) Population activity showed no self-excitation transients E) Simulation of checkerboard responses when couplings filters are those inferred from moving bars data with whole log-$\ell$ maximization. The model shows self-excitation during all runs.

## 6  Two-step inference allows for generalizing across stimuli

So far we have shown how our two-step approach can disentangle the inference of neuronal couplings from stimulus correlations. If these couplings are only due to network effects, one should expect them to generalize across stimulus conditions. To test for this, we run model simulations of one stimulus using its stimulus filter and the coupling filters inferred from the other. For the checkerboard movie (Fig. 4), and compared to the case where couplings are inferred on the same stimulus, with our two-step inference we obtained performances that are almost equal for the PSTH ($\rho = 0.81 \pm 0.05$ std, versus $\rho = 0.81 \pm 0.05$ std) and rather good for noise correlations (CoD= 0.84, versus CoD= 0.95). In addition, we never observed self-excitation (Fig. 4D). By contrast, when we used the couplings inferred by whole log-$\ell$ maximization, self-excitation happens so often (93% of the time in 100% of the repetitions) that we were not able to estimate the model performance (Fig. 4E).

For the moving bars video (Fig. S2), our two-step inference yielded performances similar to the case where couplings were inferred on the same stimulus (Table 1). Using the couplings inferred by whole log-$\ell$ maximization instead, the model performance decreased for the PSTH ($\rho = 0.65 \pm 0.12$ std, versus $\rho = 0.71 \pm 0.10$ std), and improved for noise correlations (CoD= 0.80, versus CoD= 0.55). In conclusion, two-step outperforms whole log-$\ell$ maximization on both stimuli (Table 1).

## 7  Deep GLM outperforms previous approaches

Our two-step inference decomposes the model training into two independent components, one for the stimulus processing and one for network effects. In the previous experiments we still used a linear convolution to process the stimulus, but thanks to this decomposition, we can also consider any machine capable of predicting the neurons firing rates $\{\lambda^i(t)\}_{i=1}^{N}$. In order to predict the response to checkerboard stimulation with higher accuracy, we inferred a deep, time-distributed CNN, a special case of CNNs [10] with the additional constraint that the weights of the convolutional layers are shared in time [25]. In our architecture, two time-distributed convolutional layers are followed by

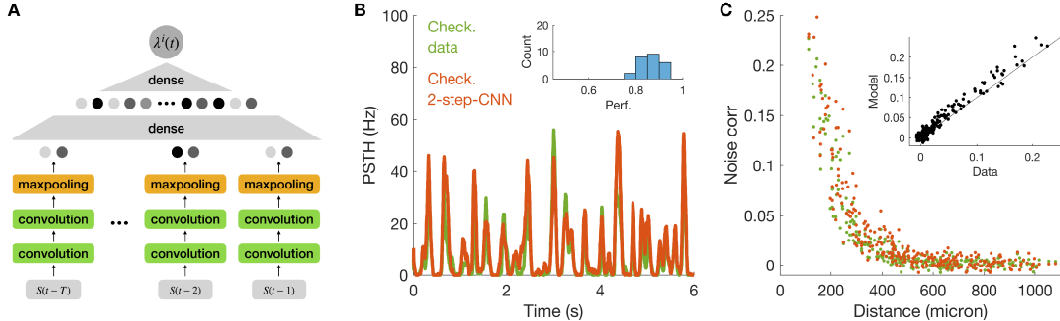

Figure 5: **Deep CNN can be included in our two-step approach to improve model performance** A) Architecture of our deep, time-distributed CNN. B) PSTH prediction for the response of an example cell to checkerboard stimulation. Inset: histogram of model performance for all cells. C) Empirical and model predicted noise correlations versus distance between cells. Inset: scatterplot.

| | Checkerboard stimulus | | | Moving bars stimulus | | |
|---|---|---|---|---|---|---|
| | PSTH | noise-corr. | self-exc. | PSTH | noise-corr. | self-exc. |
| whole log-$\ell$ maximization | $0.82 \pm 0.05$ | $0.94$ | $0\%$ | $0.71 \pm 0.10$ | $0.55$ | $10\%$ |
| two-step approach | $0.81 \pm 0.05$ | $0.95$ | $0\%$ | $0.72 \pm 0.10$ | $0.91$ | $0\%$ |
| coupl. exchange max log$\ell$ | unstable | unstable | $93\%$ | $0.65 \pm 0.12$ | $0.80$ | $0\%$ |
| coupl. exchange two-step | $0.81 \pm 0.05$ | $0.84$ | $0\%$ | $0.73 \pm 0.09$ | $0.91$ | $0\%$ |
| CNN | $0.87 \pm 0.04$ | $0.93$ | $0\%$ | — | — | — |

Table 1: **Model performance for different inference approaches.** We computed Pearson's correlation coefficients between empirical and model predicted firing rate (PSTH). For noise correlations, we estimated the CoD between model predictions and data. The third and forth rows refer to simulations that use the coupling filters inferred from the other stimulus.

a max-pooling and eventually by two dense layers that output the firing rate $\lambda^i(t)$ (Fig. 5A, see supplementary section 4 for more details). After training, we included the model in our two-step inference to build a model with both a deep architecture for the stimulus component, and a network of coupling filters.

The model shows higher performance in predicting the PSTH: $\rho = 0.87 \pm 0.04$ std, versus $\rho = 0.82 \pm 0.05$ std and $\rho = 0.81 \pm 0.05$ std, when compared to our previous models (Fig. .5B). In addition, the model was capable of predicting noise correlations with high accuracy (Fig. .5C, CoD= $0.93$, versus CoD= $0.94$ and CoD = $0.95$). We also did not observe any self-excitation transient. In summary, the model combines the benefits of deep networks with those of the GLM with its neuronal couplings.

We summarise all the different model performances in Table 1.

## 8    Discussion

In this work we have studied the application of the GLM to the case of retinal ganglion cells subject to complex visual stimulation with strong correlations. We have shown how whole log-$\ell$ maximization over all model parameters leads to inferring erroneous coupling filters that reflect stimulus correlations (Fig. 1G). This effect introduces spurious noise correlations when the model is simulated (Fig. 1E&F), prevents its generalization from one stimulus ensemble to another (Fig. 4E), and increases the chance

of having self-excitation in the network dynamics (Fig. 1G). This last issue poses a major problem when the GLM is used as a generative model for simulating spiking activity.

To solve these issues we have proposed a two-step algorithm for inferring the GLM that takes advantage of repeated data to disentangle the stimulus processing component from the coupling network. A similar approach has been proposed in the context of maximum entropy models [26, 27], and here we have fully developed it for the GLM. Our method prevents the rise of large couplings reflecting strong stimulus correlations (Fig. 3D). The absence of these couplings lowers the probability of observing self-excitation (Fig. 3E) and the inferred GLM does not predict spurious noise correlations (Fig. 3B&C). In addition, with our two-step inference the couplings are robust to a change of stimulus, and allows for generalizations (Fig. 4). In particular we showed that a model with the stimulus filter inferred from checkerboard data but couplings inferred from moving bars stimulation predicts with high accuracy the response to checkerboard.

The strongest drawback of using our method is the requirement of repeated data, which are not necessary for whole log-$\ell$ maximization of GLM. This may limit the application of our inference approach. However we emphasize that only $165s$ of repeated data were needed for inferring the couplings. In addition, another possibility that deserves to be tested is the use of spontaneous activity instead of repeated stimuli. For the retina, this activity can be recorded while the tissue is exposed to a static full-field image (blank stimulus). However, as spontaneous activity is usually very low, these recordings need to be long enough to measure correlations with high precision.

Another important contribution of our work is the possibility to easily include deep CNNs into the GLM to increase its predicting power. Deep CNNs represent today one of the best options for modelling and predicting the mean response of sensory neurons to complex stimuli such as naturalistic ones [10, 17, 18, 19], and architectures based on deep CNNs expanded with recurrence are therefore of great interest for studying the neural dynamic of sensory systems [28]. However, building a deep network that take as input both stimulus and the past activity of the neural population can be very challenging, as it implies dealing with very heterogeneous inputs. Our framework solves this problem by separating the CNN inference from that of coupling and spike-history filters, and can thus be easily added on an already inferred CNN.

The GLM has been used to estimate the impact of correlated noise on information transmission, but mostly for stimuli with low complexity [1, 29]. Future works can apply our method to model the responses to complex stimulations and study its impact on stimulus encoding.

## Broader Impact

In this work we present a computational advance to improve the inference and performance of the GLM. As the GLM is one of the most used models in computational neuroscience, we believe that many researchers can benefit from this work to advance in their investigations. The fight against blindness, which affects about 45 millions people worldwide, is one of such possible applications. Retinal prostheses, where an array of stimulating electrodes is used to evoke activity in neurons, are a promising solution currently under clinical investigation. A central challenge for retinal implants is thus to mimic the computations carried out by a healthy retina to optimize information sent to the brain. Modeling retinal processing could thus help optimize vision restoration strategies in the long term [30].

We believe that no one will be put at disadvantage from this research, that there are no consequences of failure of the system. Biases in the data do not apply to the present context.

## Acknowledgments and Disclosure of Funding

We thank M. Chalk and G. Tkacik for useful discussion. This work was supported by ANR TRAJEC-TORY and DECORE, by the European Union's Horizon 2020 research and innovation programme under grant agreement No. 785907 (Human Brain Project SGA2), by a grant from AVIESAN-UNADEV to OM, by the French State program Investissements d'Avenir managed by the Agence Nationale de la Recherche [LIFESENSES: ANR-10-LABX-65], by the Programme Investissements d'Avenir IHU FOReSIGHT (ANR-18-IAHU-01) and by Agence Nationale de la Recherche grant ANR-17-ERC2-0025-01 "IRREVERSIBLE".

The authors declare no competing interests for this paper.

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
