[Supplementary Material]

## S1 Empirical data and correlations

Figure S1: **Stimulus and noise correlation in the retinal response** A) Mosaic for $N = 25$ OFF alpha cells. B) Scatterplot of total pairwise correlation between the spiking activity in response to checkerboard and moving bars video. C) Total pairwise correlation versus cell distance D) Stimulus correlation versus cell distance E) Noise correlation versus cell distance

Responses to checkerboard and moving bars stimuli show different correlation patterns (Fig. S1). The moving bars video induces much stronger and long-ranged stimulus correlations, especially for certain pairs of distant cells. On the contrary, noise correlations decrease smoothly with distance and are of similar magnitude in the two datasets.

## S2 Correction for the absolute refractory period

As explained in the main text, when we add the two-step coupling filters to the LNP model, we need to correct the $h_{int}^i$ by its mean, Eq.4. However this correction does not take into account the addition of an absolute refractory period. In fact, if we start with an LNP model with rate $\lambda(t)$, and we prevent the cell to spike if it has spiked in the previous $\tau_{\text{refr}}^i$ time-bins during simulations, then the model rate will become a random variable itself with an average lower than $\lambda(t)$. In order to correct for this effect, we need first to quantify the mean of $n(t)$, the spike-count at time $t$:

$$
\begin{aligned}
\mathbb{E}\left(n(t)\right) &= \mathbb{E}\left(n(t) \sim Pois(\lambda(t)) \,\Big|\, \sum_\tau n(t-\tau) = 0\right) \\
&= \mathbb{E}\left(n(t) \sim Pois(\lambda(t))\right) \text{Prob}\left(\sum_\tau n(t-\tau) = 0\right) \\
&\approx \mathbb{E}\left(n(t) \sim Pois(\lambda(t))\right) \prod_\tau \text{Prob}\left(n(t-\tau) = 0\right) \\
&= \lambda(t) \prod_\tau \exp\{-\lambda(t-\tau)\}
\end{aligned}
\tag{5}
$$

where the approximation is valid under the hypothesis of small $\lambda$. By taking the log of Eq. 5, we obtain the correction term $\sum_\tau \lambda(t-\tau)$ that needs to be added to $h_{int}(t)$ in order to correct for the addition of the absolute refractory period.

## S3 Generalization results for moving bars stimulus

Figure S2: **Generalization results for moving bars stimulus** Simulation of the moving bars responses for a model where stimulus filters were inferred from the response to moving bars and couplings filter were inferred from the checkerboard data (opposite of Fig. 4) with whole log-$\ell$ maximization (A,B,C) and with our two-step inference (D,E,F). A,D) PSTH predictions. B,E) Noise correlations. C,F) Noise cross-correlation.

## S4 Time Distributed Convolutional Neural Network

In section 7 we introduce the constrained architecture of Time Distributed Convolutional Neural Networks. In order to exploit the information in the 2D spatial structure of the data we use two convolutional layers, as it is successfully done in [10], with kernels of 8x8 and 5x5 size and two feature channels each. A MaxPooling layer of pool size 2x2 is then subsequently applied to complete the spatial computation of the network. We additionally impose a Time Distributed architecture [25], i.e. the independent application of the same spatial computation to each time slice of the input, as can be seen Fig. 5. Each temporal slice of the input is compressed though the convolutional part of the network to two real numbers. Subsequently the temporal information is combined through a dense layer of 100 units with softplus activation function. A Dropout layer is additionally implemented before the last layer in order to enforce regularisation.

This architecture reduces the number of parameters to $\approx 3000$. Each model is trained for 30 epochs using the Adam optimiser on batches of 200 samples. A validation set was used to monitor the inference.