[Reviews · NeurIPS 2020]

Review 1

Summary and Contributions: Proposes a new model fitting method to fit GLM models to neural data which overcomes 1) stimulus bias, 2) instabilities in responses, and 3) inability to model complex neural computations. The method is demonstrated on a number of datasets from rat retina.

Strengths: Very nice work! surprisingly simple and insightful analysis to address a longstanding problem with these models.

Weaknesses: having a hard time thinking of any.

Correctness: as far as I know.

Clarity: yes

Relation to Prior Work: yes

Reproducibility: Yes

Additional Feedback:


Review 2

Summary and Contributions: [UPDATE AFTER REBUTTAL] I want to thank the authors for their response. After discussion with the other reviewers I stand by my positive assessment and would like to see the paper accepted. [ORIGINAL REVIEW] The paper proposes a method to train the well-established generalized linear model (GLMs) in a way that disentangles the effects of stimulus correlations and noise correlations between neurons. This is motivated by showing that GLMs trained by maximizing the complete log likelihood fail to capture the noise correlations well. Without modifying the GLM, the proposed two-step training method learns the linear filters that process the stimuli separately from the neuron coupling filters. This approach was tested on a data set 25 rat retinal ganglion cells to white noise (checkerboard) and moving bar videos. For the moving bar videos with high stimulus correlations, the authors show that their training approach avoids diverging self-excitation of the neurons and captures noise correlation better than a GLM trained by maximizing the whole log likelihood. The authors show that the inferred coupling filters between neurons generalize better across stimulus types. Finally, the authors show that their two-step training approach can also be used with multi-layer (CNN) encoders instead of linear stimulus filters. Overall, this is a well-motivated and well-written paper that presents a clear contribution and I strongly support its acceptance.

Strengths: The paper is well motivated by describing issues of the well-known GLM model. Those issues are explained clearly and after a nice summary of the GLM model in Section 3 the issues are shown in Section 4 of this paper which makes comparison to the proposed solution, the two-step training approach, simple. Section 5 shows clear improvements in capturing the noise correlations and improved stability in terms of self-excitation. The authors convincingly show that their two-step approach leads to more generalizable neuron couplings across stimulus statistics. The figures nicely depict the issues of the GLM inferred by maximizing whole log-likelihood and the improvements by using the two-step inference approach. The work is novel and very relevant for the computational neuroscience community.

Weaknesses: The paper does not include a statement about availability of code and data. Section 7 (CNN) is missing some details on model architecture and training. Especially: How and how long was the model trained? How was the architecture selected amongst others? Are there any hyperparameters, if so, how have they been selected? What did the authors do about overfitting? Was a validation set (separate from the test set) used for hyperparameter optimization? What was the size of the convolutional kernels, how many feature channels were used? What was the size of the max pools? Specification of the dense layers, i.e. number of channels and units? In general, which activation functions were used? All of this prevents the reader from thoroughly assessing the methods and results presented in this section. The authors should make sure to specify the model as precisely as possible. The paper does not discuss whether and how stimulus filters generalize across stimuli. I wouldn’t be surprised if they do not generalize – and I don’t think it would be a problem for the paper either, but stating the result would be great.

Correctness: The paper’s exposition of the findings and the methods used seem to be sound and correct.

Clarity: In general, the paper is very well-written and structured clearly. Improvement is possible in some paragraphs: In section 5, paragraph “Full model.”, it is not clear to me why the correction Eq. (4) needs to be done. I would have thought that J_ij is learned such that it does not introduce a bias since during training h_stim is given by the log-PSTH and the authors perform maximum likelihood. The paper would benefit from clarifying why this needs to be done. Also, some details should be expressed more clearly: In equations 2, 3, 4 it should be stated over what range of values the summation index \tau runs. I assume \tau=0 is not included in the summation. In l. 100 (and elsewhere) it is not clear to me if the value for \trau_ref is inferred during training or set. If the latter is true, please specify its value and reason for choosing it. In ll. 111, 116 and elsewhere, please specify what the “+/-“ refers to exactly (e.g. standard deviation, standard error, confidence interval, …). In the Reference section, a couple of references are cited as preprints of published papers or missing the publication outlet.

Relation to Prior Work: Related work is discussed appropriately and it is clearly described how this work goes beyond existing work.

Reproducibility: Yes

Additional Feedback: Why is the analysis depicted in Fig. 3D not done/shown for the case presented in Fig. 4? Why was the experiment described in Section 7 not done on the moving bar video stimuli? How exactly is the coefficient of determination for quantification of noise correlations computed? Is it simply R^2 of a (linear?) regression model of correlation vs. distance or does it quantify how well the noise correlations between individual pairs of neurons are predicted on average? Also, if I understand it correctly, they authors quantify only the zero-lag correlation, but the largest errors appear to be happening at non-zero lags. Did the authors investigate that? Reproducibility: No indication is given whether code and data will be published. I highly encourage the authors to do so to ensure reproducibility.


Review 3

Summary and Contributions: The paper discusses an improvement to Generalized Linear Models. The authors proposes a two step training approach. In the first step, the stimulus related parameters are inferred by training lNP model while ignoring the internal coupling from unrepeated stimulus responses. In the second step, the authors utilize repeated inputs to compute PSTHs and use the neural responses to infer the GLM coupling parameters. The authors apply the this approach to rat retina recordings in response to visual stimuli.

Strengths: This work proposes a new training procedure for GLMs. This allows developing better brain models. Further, the combination of CNN models with GLMs could help guide the search of better architectures for computer vision.

Weaknesses: In the first step, the network coupling and internal dynamics are ignored as the system is assumed to be completely driven by the stimulus. In reality, I believe the internal dynamics may still play a role. I think more clarification is needed for why this does not matter. Despite the two step approach seems to show advantages, I wonder if another step of fine tuning the whole model after the two step training procedure could provide further improvements. In reality, I think both stimuli and dynamics of the network shapes the responses. So, my intuition is that training all components of the model, which matches better the real setting, but starting from a good initialization obtain by the two step approach could potentially yield better models. This idea also links to pretraining in neural network, where a pretraining procedure followed by fine tuning may outperform pretraining steps alone or training the model alone. Post rebuttal phase: I think this is an interesting work. I read the author response and they adequately address the points I raised.

Correctness: The claims and methodology seems correct.

Clarity: The paper is clearly written. Some points to improve clarity: - In the last paragraph in the intro, the two steps are presented in the reverse order to Section 5, which could be confusing to the reader. - More clarity an explanations are needed for the first limitation presented in the second paragraph in the intro. - I think the approach of using Conv nets with GLM in Section 7 is very interesting. I think expanding this section more and running more experiments discussing whether the model performance could be enhanced by using different architectures (eg deeper models/ resnet blocks etc) could largely improve the impact of this paper. In particular, could this approach be used to discover more biologically plausible architectures by comparing the fit of GLM models with different convolution architectures to neural recordings and ranking models in their match.

Relation to Prior Work: I think the paper adequately describe prior work.

Reproducibility: Yes

Additional Feedback:


Review 4

Summary and Contributions: The authors propose a method to prevent runaway excitation in GLM models of spiking networks. Further, they show how to incorporate the benefits of deep network models with GLMs.

Strengths: The paper tackles a concrete problem and does a good job of citing prior work to show that this is an issue with GLMs that people care about.

Weaknesses: The reason why the method works to prevent runaway excitation is not rigorously proven or explained particularly well. Also, a few past approaches (e.g., Gerhard et al., 2016) are mentioned but not explicitly compared against on the experimental data.

Correctness: Yes.

Clarity: The paper is reasonably well-written, but some explanations and intuitions about the modeling approach are missing (see below).

Relation to Prior Work: References 12-14 were useful citations that helped me understand the problem the authors were able to solve. While these references are briefly mentioned, they are not revisited later in the paper and I am not sure why the authors' proposed method is better than, for example, reference 14 (Gerhard et al., 2013) which also proposes a seemingly reasonable way to prevent runaway excitation. The authors say that this existing approach is "sub-optimal" but they do not compare their results directly to this past work and thus it is hard for me to evaluate that claim. Presumably the two-step approach proposed by the authors is also "sub-optimal" in some sense -- from the standpoint of maximizing log-likelihood the "optimal" method is "whole-log-likelihood maximization" (but of course this is what leads to runaway excitation!). The authors claim that an advantage of GLMs over deep network models is that they can account for temporal correlations between neurons through coupling filters. Arguably this has been incorporated into certain models via recurrent layers -- https://arxiv.org/abs/1807.00053 -- I think the authors should be a bit more careful in their statement. I think it's fair to say that GLMs account for Poisson spiking statistics (unlike most deep networks) while also introducing temporal dependencies. I'm not sure this is any fundamental limitation of the deep network approach though.

Reproducibility: Yes

Additional Feedback: - The authors claim that empirically they do not need large amounts of repeated stimuli for the method to work. This empirical claim is based on only a single experimental dataset. It would be nice to see some theoretical analysis or exploration into how much data is needed for this to work -- presumably if my data has only 2 repeats of a stimulus then the h_stim auxilliary variable could be very poorly estimated. This introduces a bias into the results of the model, but how bad is this bias? - The correction terms (e.g. equation 4) seem to be somewhat ad hoc heuristics. Is this correction procedure provably optimal in some way? What is the overall objective function that is optimized by the two step procedure? ---- ADDITIONAL FEEDBACK AFTER REBUTTAL ---- Based on the comments from other reviewers and the author feedback I've increased my score to a 6, and decreased by confidence to a 3. Coming from a somewhat outside perspective (I've read plenty of literature on these sensory encoding models, but my research interests are mostly elsewhere), the motivations of this paper and the advantages of the modeling approach were somewhat difficult for me to weigh. The two-step approach seems somewhat ad hoc, ideally the model would be trained end-to-end --- I mentioned this in my original comments asking "what is the overall objective function" that is optimized by the two-step approach? Why is the correction term in equation 4 justified and is it provably optimal? (see also reviewer 3's related comments on fine tuning). These weren't fully addressed in the rebuttal. I regret not emphasizing some of these points more strongly in my original review. I may have gotten hung up on the specific issue of runaway excitation in my review, but the authors do heavily emphasize it as a major advance of their work. It still isn't totally clear to me that existing approaches to fixing this runaway excitation problem wouldn't also have positive benefits on estimating noise correlations more generally (the two problems seem related to me, but perhaps my intuition is wrong here). I think this points to problems in the clarity and presentation of the manuscript --- if the runaway excitation is indeed a tangential or minor point, the writing should be edited to reflect this. Overall though, I'm happy that the authors will add more details / citations regarding recurrent layers in deep network models. I also think that the basic idea of adding repeated stimuli is a good one. Further, the authors do provide a proof-of-principle that this idea may be incorporated and extended into training deep networks. The enthusiasm from the other reviewers also suggests that these ideas will be of greater interest than I would have guessed. Some of my reservations about the paper relate to writing and presentation --- these issues are addressable during the revision process and the authors seem receptive to adding additional clarity.

[Author Response · NeurIPS 2020]

**To all reviewers.** We thank the reviewers for the positive evaluation of our work and for all the suggestions for improving the clarity of our manuscript. Firstly, let us first state clearly that, upon acceptance, we will happily share our code and data. Many reviewers' concerns focused on the correction procedure of Eq. 4. This correction is necessary because when we learn the stimulus filters (step-1) the couplings are set to zero. However, when we include them in the *full model* step, they generate a mean contribution to the PSTH that needs to be subtracted. Otherwise this contribution would sum up to the stimulus signal and cause an over-estimation of the PSTH. More intuitively, and as reviewer 3 points out, the internal dynamics and so the coupling network can also drive the system, and in the step-1 of our procedure we neglect this. This is exactly why we need to introduce the correction of Eq. 4, which accounts for the contribution of the network to the mean response (PSTH). In addition, the correction term of Eq. 4 can be seen as a first order approximation of an optimal Plefka expansion (Plefka 1982, Kappen and Rodirguez 1997) around vanishing couplings. It could be possible to extend our method to second (called TAP) or higher orders. However, as we did not observe significant improvements on our results, we decided to keep the first order only, and to present it as a simpler correction accounting for the mean effect. Note that Eq. 4 just subtracts the mean of the left hand side. We will take care of explaining all this better.

**Reviewer 1.** Thanks, we have nothing to add.

**Reviewer 2.** The reviewer's major concern is about our CNN example (Sect. 7). Its scope within the paper is to showcase the possibility of including deep architectures within our GLM framework. Deep CNNs have already been shown to have great performance on retinal recordings (Ref. 9), and it is not our scope to deepen these investigations, nor to claim that our bare CNN (excluding the GLM coupling filters) is better than those already presented in the literature. We will anyhow add a supplementary section with more explanation and details about our CNN model. For the same reason, we did not find it necessary to develop a whole CNN model for the moving bar that would have distracted the reader from our main messages. Minor points. As the reviewer supposes, stimulus filters do not generalise across stimuli, neither are they expected to do so. They are even of different dimensions: 3d for white noise, 2d for moving bar. We acknowledge that one can misunderstand this from the abstract, and we will modify it promptly. In addition, the analysis of Fig.3D is not done for the case of Fig.4 because when simulating the model with the couplings inferred via maximum log-$\ell$ (as opposed to our strategy) we observed self-excitation during all runs (Fig. 4E), and this prevented us from estimating predicted noise correlations. Lastly, for evaluating our prediction of noise correlations, the coefficient of determination (CoD) i is computed as 1- var(error) / var(data), and not as $\rho^2$. We will add these explanations.

**Reviewer 3.** Additional fine tuning. Good point. The correction of Eq. 4 plays the role of an additional fine tuning after the separate inference of stimulus and couplings filters. As explained above, Eq. 4 is indeed an approximation. An additional, not approximated fine tuning could consist in an inference of the stimulus filters after freezing the couplings network from step-2. We tried, but this approach came with no improvements and at a cost of more complex and time-consuming inferences. We therefore avoid including this additional step. We will add a comment in the discussion. Finally, we agree that including more analysis on deep architectures within the GLM framework opens for novel and interesting investigation. This is what we plan for future works, but it lies beyond the scope of the present work.

**Reviewer 4.** We believe the reviewer is missing a large part of our results, by focusing their criticisms on runaway instabilities. While important, these instabilities are not the main focus of our manuscript. In addition to runaway effects and the inclusion of CNNs into GLMs, we have focused on preventing stimulus biases in the GLM inference (Sect. 4, 5 and 6). These biases are particularly detrimental for complex stimuli: they prevent the correct estimation of noise correlations (Fig. 1G&E), and reduce the generalisability of inferred parameters (Fig. 4E) . Our two-step inference is constructed for dealing with these problems and is shown to solve them (Figs. 3 and 4). It accounts **also** for runaway effects, showing that the model inferred with our strategy is robust and powerful. Prior works. Gerhard et al. (2017) focuses only on single-neuron GLMs and lets for future developments the case of GLMs with a coupling network. That work surely merits a citation, but a quantitative comparison with our method for neuronal populations would not be meaningful. In addition, we never claim that our method provides better results than the other discussed approaches (Refs. 12 and 13), but simply that they are computationally harder than our method. Recurrent CNN. We agree that CNNs equipped with recurrent layers can account for temporal noise correlations between neurons. In fact, in a broad sense our method falls in this class of models. In the text we were referring to non-recurrent CNNs, yet we acknowledge the reviewer's criticism and we will happily add a comment and some citations to avoid additional misinterpretations. Repeated data. As clearly stated in the discussion, we acknowledge that the necessity of repeated data is a strong limitation of our method. For all the computational experiments we present we used only 2/3 mins of recordings, and this was somehow the lower-bound for our approach to work. However, let us stress that 2/3 mins does not represent a prohibitive cost within experiments that can last for two hours at least.

[Meta-Review · NeurIPS 2020]

This paper presents a novel methodology to fit generalized linear models to neural data, overcoming the various limitations of existing models which are prevalent in the literature. The paper received 4 thoughtful and thorough reviews. There was significant discussion following the author response. One reviewer found that the authors did not provide significant intuition or evidence why their method prevents "runaway excitation". However, the other three reviewers believed this was secondary and argued strongly for acceptance, considering this a significant advance in GLM modeling of neural data.